# The Effect of *Atm* Loss on Radiosensitivity of a Primary Mouse Model of *Pten*-Deleted Brainstem Glioma

**DOI:** 10.3390/cancers14184506

**Published:** 2022-09-17

**Authors:** Connor E. Stewart, María E. Guerra-García, Lixia Luo, Nerissa T. Williams, Yan Ma, Joshua A. Regal, Debosir Ghosh, Patrick Sansone, Mark Oldham, Katherine Deland, Oren J. Becher, David G. Kirsch, Zachary J. Reitman

**Affiliations:** 1Department of Radiation Oncology, Duke University Medical Center, Durham, NC 27710, USA; 2Department of Pediatrics, Mt. Sinai Hospital, New York, NY 10029, USA; 3Department of Pharmacology & Cancer Biology, Duke University Medical Center, Durham, NC 27710, USA; 4Department of Pathology, Duke University Medical Center, Durham, NC 27710, USA; 5Department of Neurosurgery, Duke University Medical Center, Durham, NC 27710, USA; 6The Preston Robert Tisch Brain Tumor Center, Duke University Medical Center, Durham, NC 27710, USA

**Keywords:** DIPG, diffuse midline glioma, ATM, radiation therapy, genetically engineered mouse models

## Abstract

**Simple Summary:**

Brainstem gliomas are deadly childhood brain tumors and new treatments are needed. The treatment meeting the standard of care is radiation therapy, but the tumors inevitably progress at some point after this treatment. One treatment strategy under preclinical investigation for brainstem gliomas is to inhibit the function of ataxia–telangiectasia mutated kinase (ATM), a master regulator of the cellular response to radiation therapy. ATM inhibition makes some tumors more sensitive to radiation therapy, but this may depend on the genetic makeup of the tumor. An important subset of brainstem gliomas harbor mutations in the *PTEN* gene or related genes. Here, we develop a genetically engineered mouse model of *Pten*-mutated brainstem glioma. We use genetic tools to test if ATM inactivation can enhance the efficacy of radiation therapy in this *Pten*-mutated brainstem glioma model. We find that ATM inactivation does not enhance the efficacy of radiation therapy for this model of *Pten*-mutated brainstem glioma. These findings indicate that *PTEN* mutational status should be considered in the design of inclusion criteria and correlative studies for future clinical trials of ATM inhibitors in brainstem glioma patients.

**Abstract:**

Diffuse midline gliomas arise in the brainstem and other midline brain structures and cause a large proportion of childhood brain tumor deaths. Radiation therapy is the most effective treatment option, but these tumors ultimately progress. Inhibition of the phosphoinositide-3-kinase (PI3K)-like kinase, ataxia–telangiectasia mutated (ATM), which orchestrates the cellular response to radiation-induced DNA damage, may enhance the efficacy of radiation therapy. Diffuse midline gliomas in the brainstem contain loss-of-function mutations in the tumor suppressor *PTEN*, or functionally similar alterations in the phosphoinositide-3-kinase (PI3K) pathway, at moderate frequency. Here, we sought to determine if ATM inactivation could radiosensitize a primary mouse model of brainstem glioma driven by *Pten* loss. Using Cre/loxP recombinase technology and the RCAS/TVA retroviral gene delivery system, we established a mouse model of brainstem glioma driven by *Pten* deletion. We find that *Pten*-null brainstem gliomas are relatively radiosensitive at baseline. In addition, we show that deletion of *Atm* in the tumor cells does not extend survival of mice bearing *Pten*-null brainstem gliomas after focal brain irradiation. These results characterize a novel primary mouse model of *PTEN*-mutated brainstem glioma and provide insights into the mechanism of radiosensitization by ATM deletion, which may guide the design of future clinical trials.

## 1. Introduction

Brainstem gliomas are incurable pediatric brain tumors for which new therapies are urgently needed. Brainstem gliomas cannot safely undergo gross total resection because of their location in an eloquent region of the brain. The only treatment that has been shown to improve chances of survival for children with brainstem gliomas is radiation therapy (RT), but the tumor inevitably progresses after treatment and almost always leads to death within 18 months. Many pediatric brain tumor clinical trials under development seek to rationally match investigational therapies with vulnerable tumor genotypes. Thus, to design future brainstem glioma clinical trials there is an urgent need to rationally identify which molecular subtypes of brainstem gliomas are likely to respond to emerging investigational therapeutic strategies.

Brainstem gliomas contain therapeutically important alterations in *Phosphatase and tensin homolog* (*PTEN)* and its molecular pathway, and frequent mutations in *TP53*. *PTEN* is a tumor suppressor that is recurrently inactivated by focal or chromosomal deletions of chromosome 10 and/or loss-of-function mutation in 6–27% of diffuse midline gliomas [1,2,3,4,5,6]. PTEN functions in the phosphoinositol-3-kinase (PI3K) molecular pathway. Mutations in other nodes of the PI3K molecular pathway also occur in a mutually exclusive fashion with *PTEN* alterations, with >20% of brainstem gliomas containing alterations in the PI3K pathway (predominantly *PIK3CA*, *PIK3R1*, and *PTEN* alterations) [1,6,7]. PI3K pathway alterations are uniquely enriched in specific anatomic and genetic subsets of brainstem glioma, indicating that they may represent a biologically distinct subset of tumors. PI3K/mTor pathway alterations have been associated with tumor location in the brainstem [1] and with therapeutically important *ACVR1* mutations [5]. Also, 17 of 37 (46%) cases of the important subgroup of H3.1K27M-mutated brainstem gliomas harbored PI3K alterations in a large pediatric brainstem glioma and non-brainstem glioma dataset [1]. Additionally, we identified an association between *PIK3CA* alterations and *TP53*-wild-type tumors with mutations in the oncogenic phosphatase *PPM1D* [6]. In addition to moderately frequent PI3K pathway alterations, up to 70% of brainstem gliomas are defined by mutations of the tumor suppressor *TP53* [1,3,6]. Identifying therapeutic approaches that may be effective in *PTEN-* and/or *TP53*-altered brainstem gliomas may therefore help guide the design of clinical trials that match investigational therapies with tumor genotypes.

Targeting the kinase ataxia–telangiectasia mutated (ATM) in combination with RT has emerged as a promising therapeutic strategy for brain tumors, and may be especially effective for brainstem gliomas harboring *TP53* alterations. Following RT, ATM is activated in response to DNA double-strand breaks to mediate the DNA-damage response that facilitates cell cycle arrest, DNA repair, and apoptosis in a cell-type-dependent manner [8]. Loss of functional ATM preferentially radiosensitizes (i) proliferative tissues, including tumors [9] and (ii) cells that have acquired mutations in the tumor suppressor p53 [10]. Loss of ATM prevents radiation-induced apoptosis of the developing brain [11,12]. These data suggest that pharmacological inhibition of ATM during RT may result in a wide therapeutic ratio for p53-mutant tumors with limited toxicity to adjacent normal brain tissue. We previously showed that deletion of *Atm* within tumor cells in a primary mouse model of p53-deficient brainstem glioma significantly improved radiation response and overall survival of mice following RT [13]. In contrast, deletion of *Atm* within tumor cells in a p53-wild-type model of brainstem glioma driven by loss of *Ink4a/ARF* did not improve radiation response or overall survival of mice following RT [13]. These results raise the possibility that inhibition of ATM in combination with RT may be an effective approach for treating brainstem gliomas with specific genotypes, such as p53-mutant tumors. Indeed, a potent ATM inhibitor that penetrates the blood–brain barrier has now been developed [14]. Such inhibitors have now entered clinical trials testing concurrent ATM inhibition and RT in adult patients with glioblastoma and brain metastases (NCT03423628). While these preclinical data nominate *TP53* status as a potential biomarker of sensitivity to ATM-directed radiosensitization strategies, it is unknown whether *PTEN*-altered brainstem gliomas are likely to respond to ATM inactivation in this manner. Because ATM is a serine/threonine kinase within the PI3K-like kinase (PI3KK) family, *PTEN*-altered tumors with activated PI3K could theoretically promote resistance to ATM inhibition.

Here we used genetically engineered mice to determine whether brainstem gliomas harboring *PTEN* loss-of-function alterations can be radiosensitized by deletion of ATM. Specifically, we generated a primary mouse model of brainstem gliomas driven by *Pten* loss and examined whether ATM inactivation can cause radiosensitization. In this model driven by *Pten* loss, in which p53 is wild-type, we find that deletion of *Atm* in the tumor cells does not affect tumor onset or aggressiveness. After characterizing an image-guided focal brainstem irradiation plan and fractionation scheme, we demonstrate that RT can lengthen overall survival for mice bearing *Pten*-null brainstem gliomas. However, we find that *Atm* deletion in tumor cells does not extend survival of mice bearing *Pten*-null brainstem gliomas after treatment with focal brainstem irradiation. These results provide insights into the mechanism of radiosensitization by ATM deletion and the therapeutic vulnerabilities of brainstem gliomas with PI3K pathway alterations. Moreover, these results suggest that defining tumor genotype will be important to interpret the activity of ATM inhibitors in future clinical trials.

## 2. Materials and Methods

Mouse strains. All mouse strains in this study have been described previously, including Nestin^TVA^, Pten^FL^, and Atm^FL^ [15,16,17,18]. *Nestin^TVA^* mice were provided by Oren Becher (Mount Sinai, New York, NY, USA). *Atm^FL^* mice were provided by Frederick Alt (Boston Children’s Hospital, Boston, Massachusetts, USA). Pten^FL^ mice were obtained from Jackson Laboratories [18]. Mouse strains were maintained on a mixed genetic background. Male and female littermate controls that retained expression of one ATM allele or lacked both alleles in their tumors were used to minimize the effect of differences in environment, sex, and genetic background. All mice used in this study are summarized in Table 1.

**Table 1 cancers-14-04506-t001:** Mice used in this study.

Genotype	End Point	No. Mice	Figure
nPtenA-FL/+	Time to tumor detection (This group includes mice that, upon tumor detection, were subjected to tissue analyses or survival experiments below)	45	Figure 1D
nPtenA-FL/FL	Time to tumor detection (This group includes mice that, upon tumor detection, were subjected to tissue analyses or survival experiments below)	56	Figure 1D
nPtenA-FL/+	Brain tissue collected within 24 h of tumor detection via in vivo imaging for HE, HA IHC, PTEN IHC, Ki67 IHC	3	Figure 1C and Figure 2B–E
nPtenA-FL/FL	Brain tissue collected within 24 h of tumor detection via in vivo imaging for HE, HA IHC, PTEN IHC, Ki67 IHC	3	Appendix A and Figure 1C
nPtenA-FL/+	Brain tissue collected 1 h after 10 Gy × 1 delivered within 24 h of tumor detection via in vivo imaging and subjected to pATM IHC, pKAP1 IHC	3	Figure 1A,B
nPtenA-FL/FL	Brain tissue collected 1 h after 10 Gy × 1 delivered within 24 h of tumor detection via in vivo imaging and subjected to pATM IHC, pKAP1 IHC	4	Figure 1A,B
nPtenA-FL/+	Survival after tumor detection without brain treatment	14	Figure 3F
nPtenA-FL/+	Survival after 10 Gy × 3 delivered following tumor detection *	17	Figure 3F
nPtenA-FL/+	Survival after 10 Gy × 3 delivered following tumor detection *	18	Figure 4
nPtenA-FL/FL	Survival after 10 Gy × 3 delivered following tumor detection	22	Figure 4

* These nPtenA-FL/+ groups shown in survival curves in Figure 3F and Figure 4 include a subset of overlapping mice. Each survival curve shows only those mice that are from the same litters as mice from the other group in the same figure (i.e., littermate-controlled).

**Figure 1 cancers-14-04506-f001:**
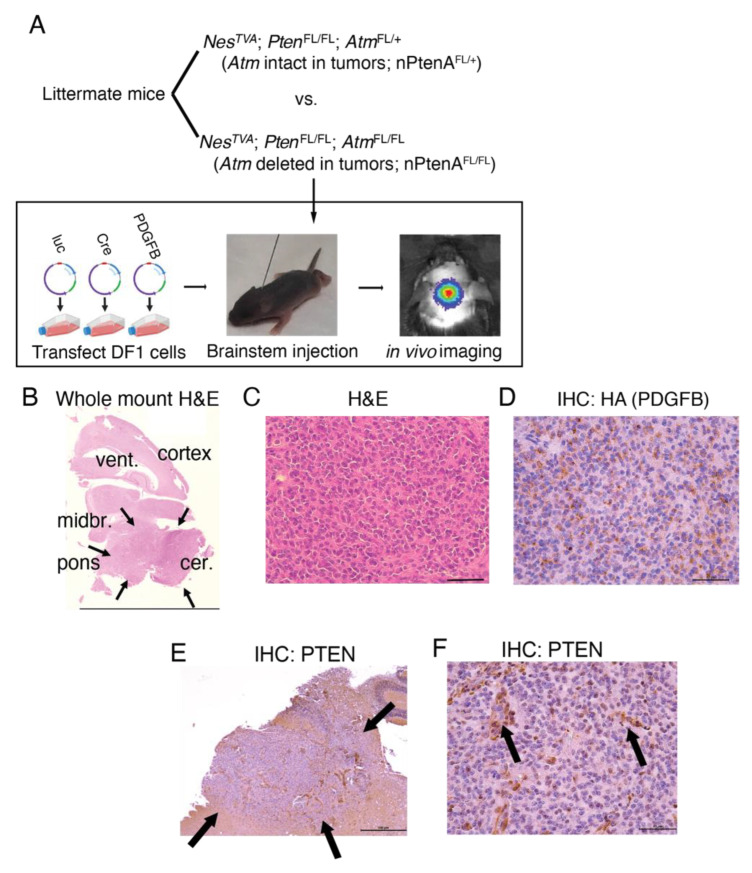
Primary brainstem gliomas lacking *Pten*. (**A**) Genotypes of mice used in this manuscript, and schematic showing workflow of generating retroviral RCAS vectors carrying Cre, luc, and PDGFB-HA payloads in chicken DF1 cells, injection of DF1 cells into the mouse brainstem, and then monitoring for tumor formation with in vivo bioluminescence imaging. (**B**) Whole-mount HE slide showing expansile tumors in the brainstem of nPtenA^FL/FL^ mice. Midbr., midbrain; vent, lateral ventricle; cer., cerebellum. (**C**) Magnified HE slides for tumors from nPtenA^FL/FL^ mice. Scale bar represents 50 μm. (**D**) Immunohistochemistry staining for HA-tagged PDGFB in tumors from nPtenA^FL/FL^ mice. Scale bar represents 50 μm. (**E**) 5× mount showing immunohistochemistry for PTEN of tumors centered in the brainstem (pons) with lack of PTEN reactivity, and diffuse infiltrating borders (arrows) for tumors from nPtenA^FL/FL^ mice. Scale bar represents 500 μm. (**F**) 40× mount shows complete loss of PTEN in tumor cells but not normal blood vessels (arrows) in nPtenA^FL/FL^ mice. Scale bar represents 50 μm.

**Figure 2 cancers-14-04506-f002:**
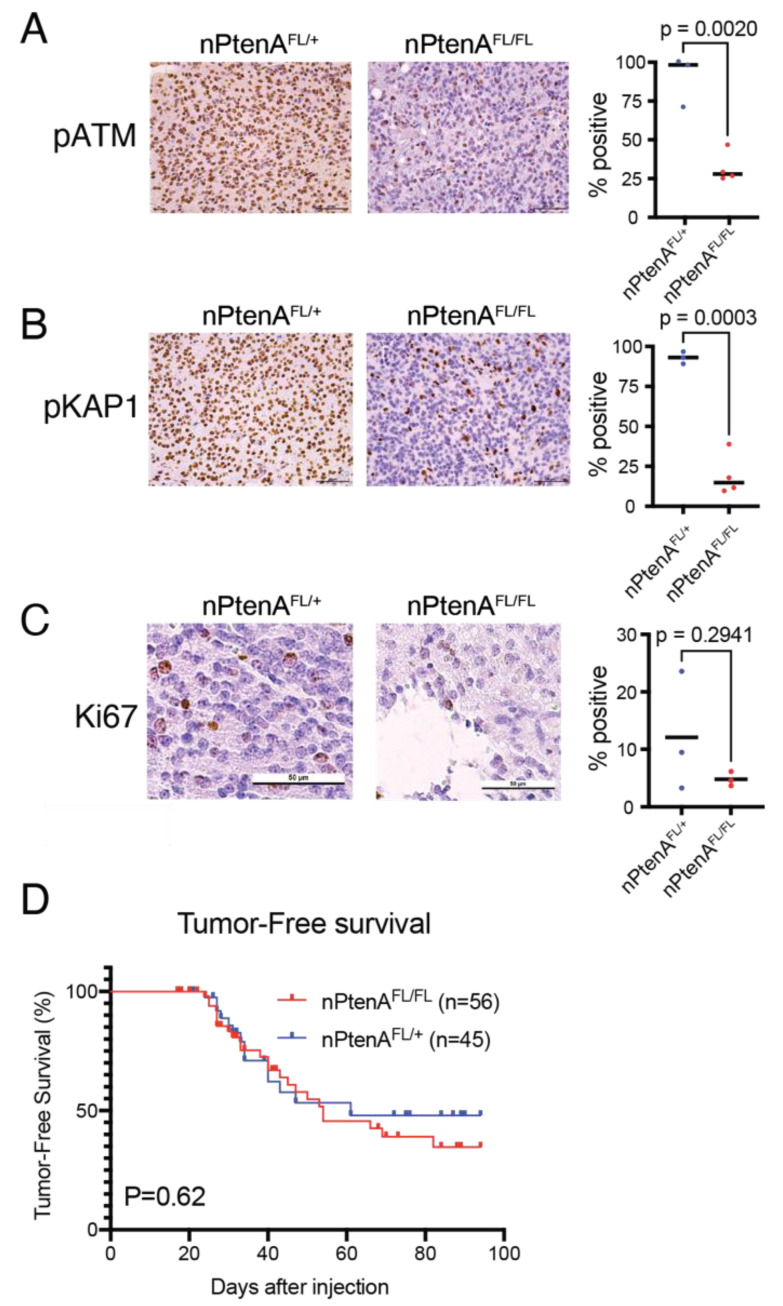
Effects of ATM loss on brainstem gliomas driven by *Pten* deletion. (**A**) Representative immunohistochemistry of phospho-Atm protein 1 h after 10 Gy focal brain irradiation in nPtenA^FL/+^ (n = 3) mice (**left**), as compared to nPtenA^FL/FL^ (n = 4) mice (**right**). Cell positivity percentage is shown on the plot on the right, and Student’s t-test *p*-value is shown. Scale bars represent 50 μm. (**B**) Representative immunohistochemistry of phospho-KAP1 1 h after 10 Gy focal brain irradiation in nPtenA^FL/+^ (n = 3) mice (**left**), as compared to nPtenA^FL/FL^ (n = 4) mice (**right**). Cell positivity percentage is shown in the plot on the right, and Student’s t-test *p*-value is shown. Scale bars represent 50 μm. (**C**) Ki67 staining in unirradiated brainstem gliomas from nPtenA^FL/+^ (**left**) and nPtenA^FL/FL^ (**right**) mice. Cell positivity percentage is shown in the plot on the right, and Student’s t-test *p*-value is shown. Scale bar represents 50 μm. (**D**) Tumor-free survival in the absence of irradiation in nPtenA^FL/+^ and nPtenA^FL/FL^ mice. *p*-value is for the log-rank test.

**Figure 3 cancers-14-04506-f003:**
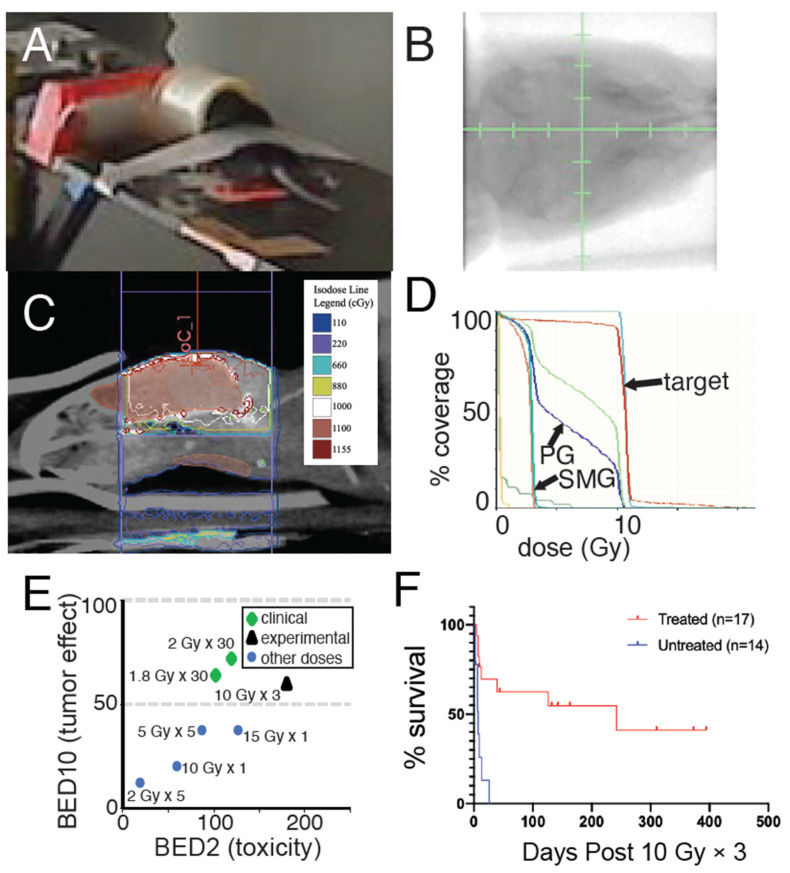
Volumetric characterization of radiation treatment plan for image-guided focal brainstem mouse irradiation. (**A**) Example of mouse anesthetized with isoflurane gas on the irradiator couch. (**B**) kV image guidance prior to radiation therapy. (**C**) Sagittal image from a cone-beam CT with isodose lines showing lateral beams targeting brainstem. (**D**) Dose-volume histogram demonstrates >95% of brainstem target is covered by radiation dose prescription with minimal radiation dose to the parotid and submandibular glands (PG, SMG). (**E**) Biologically effective dose (BED) calculations identify 10 Gy × 3 = 30 Gy (

 ) as biologically similar to the standard-of-care for DMG of 1.8 Gy × 30 = 54 Gy (

 ) compared to other fractionation schemes ( 

). BED10 is shown on the y-axis to model effective dose to rapidly dividing tissues, such as tumors. BED2 is shown on the x-axis to model effective doses for slowly dividing normal tissues, such as normal brain tissue, estimated to have a low α/β ratio of two. (**F**) Survival of nPtenA^FL/+^ mice with tumors lacking PTEN but retaining ATM treated with or without 10 Gy × 3. *p* < 0.001, log-rank test.

**Figure 4 cancers-14-04506-f004:**
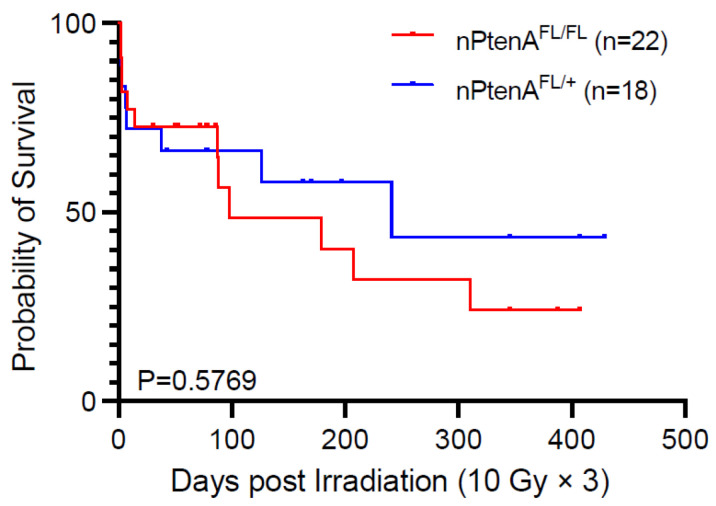
Effect of ATM loss in tumor cells on survival of mice bearing *Pten*-null brainstem gliomas after irradiation. Survival of nPtenA^FL/FL^ mice with tumors lacking ATM following 10 Gy × 3 of focal brain irradiation is compared to littermate control nPtenA^FL/+^ mice with tumors that retain *ATM*. *p*-value is for the log-rank test.

Chicken fibroblast cell culture and RCAS retrovirus generation. Primary murine brainstem gliomas were induced via RCAS viral gene delivery. Chicken fibroblast DF1 (ATCC CRL-12203) cells were transfected with plasmids for RCAS-Cre, RCAS-luciferase, and RCAS-PDGFB using X-tremeGENE 9 following the manufacturer’s instructions. The cells were cultured at 39 °C in DMEM with 10% fetal bovine serum, 2 mM L-glutamate, and 1% penicillin-streptomycin. Virus-producing DF1 cells were injected into the brainstem of *Nestin^TVA^* mouse pups at postnatal day 3–5 using a Hamilton syringe. The various RCAS viruses were injected in equal proportion, totaling 1 × 10^5^ cells per mouse, suspended in 1 µL of media.

Bioluminescence imaging. Infection of Nestin^TVA^ mice with RCAS-Luciferase enabled the use of in vivo bioluminescence imaging to identify mice with primary brainstem gliomas. Mice were injected intraperitoneally with 150 mg/kg D-luciferin, which is converted to oxyluciferin by the firefly luciferase enzyme in a reaction that emits 560 nm light. Luciferase activity was monitored weekly beginning 3 weeks post-tumor initiation using the IVIS Lumina III (PerkinElmer).

Immunohistochemistry. Whole mouse brains were harvested either one hour after radiation therapy or upon presentation of neurological symptoms. Brains were fixed in 10% formalin for 24–72 h, then stored in 70% ethanol until histological preparation. The brains were embedded in paraffin wax, and sagittal sections were cut for subsequent analysis.

For histological and molecular analysis, the brain sections were subjected to hematoxylin and eosin staining and immunohistochemistry. Following tissue rehydration, endogenous peroxidase activity was neutralized with 3% hydrogen peroxide (Millipore Sigma). The slides were treated with Antigen Unmasking Solution (Vector Laboratories) and heated in a microwave oven at low power. Normal serum with 0.25% Tween 20 (Vector Laboratories and Millipore Sigma) was used for blocking. The slides were incubated overnight with the following primary antibodies at the indicated dilutions: rabbit monoclonal anti-HA 1:1000 (Cell Signaling Technology 3724), rabbit polyclonal anti-PTEN 1:400 (Thermo Fisher 51–2400), rabbit polyclonal anti-KAP-1(phospho-S824) 1:200 (Bethyl Laboratories A300-767A), and rat monoclonal anti-mouse Ki67 1:200 (BioLegend 652402). Mouse monoclonal anti-human-ATM(phospho-S1981) which stains the homologous mouse ph-ATM(phosphor-S1987) was used at 1:500 (Abcam ab36810). The slides were incubated in biotin-conjugated secondary antibodies, then staining was visualized using VECTASTAIN Elite ABC-HRP Reagent followed by incubation with DAB Substrate Kit (Vector Laboratories). The slides were counterstained with Mayer’s hematoxylin (Millipore Sigma) and dehydrated using a gradient of ethanol and water solutions. Stained slides were imaged with a Leica DFC450 bright-field microscope using Leica Suite software. For pATM, pKAP1, and Ki67 quantification, a random tissue area within the tumor was selected, and the percent of cells stained positive was recorded by a single observer who was blinded to the genotype and treatment.

Image-guided focal brainstem irradiation and 3D planning. After detecting brainstem gliomas through bioluminescence imaging, we treated mice with three daily 10 Gy fractions of focal brain irradiation using an X-RAD 225C× small animal image-guided irradiator (Precision X-Ray). The brains were irradiated with lateral fields using a 15 × 20 mm rectangular radiation field with an average dose rate of ~280 cGy/min. Mice were anesthetized with 2–3% isoflurane for all procedures. Three-dimensional radiation plans were reconstructed using cone-beam CT data acquired on a Small Animal Radiation Research Platform (SARRP, Xstrahl) using the same anesthesia, immobilization, and irradiation setup described above. Volumetric treatment planning and dose-volume histogram generation was accomplished in Muriplan 3.0.


Biologically effective dose calculations.


The formula
BED = n × d [ 1 + d/(α/β) ]
was used, where n is the number of treatment fractions, d is dose per fraction in Gy, and α/β is the dose at which the linear and quadratic components of cell kill are equivalent. BED10 reflects BED for α/β of 10, and BED2 reflects BED for α/β of 2.

Statistics. Analysis of Kaplan–Meier curves for tumor-free survival and overall survival studies was performed using the log-rank test. A two-tailed Student’s t-test was used to test for differences of immunohistochemistry scoring between groups. To detect a hazard ratio of 3, 34 deaths need to be observed with 90% power using a two-tailed log-rank test (α = 0.05), assuming 10% censoring, an 80% baseline event rate at 100 days, and 300 days of follow up. Therefore, for experiments we aimed to randomize 38 mice (19 per genotype) and follow them for neurological decline when mice were euthanized.

Study approval. All animal studies were approved by the Duke Institutional Animal Care and Use Committee.

## 3. Results

### 3.1. Establishment of Brainstem Gliomas Lacking Pten and ATM

Our workflow to generate primary brainstem gliomas driven by *Pten* deletion, with and without tumor cell-specific deletion of *Atm*, is shown in Figure 1A. The avian retrovirus RCAS was used to regulate the expression of relevant genes in progenitor cells of the mouse brainstem. To infect brain progenitor cells with RCAS, we used mice that had been engineered to express the RCAS cognate tumor virus A (TVA) receptor in cells that express Nestin (Nestin^TVA^). Chicken fibroblast cells expressing various RCAS viruses were injected into the brainstem of neonatal mice. RCAS was used to introduce expression constructs for luciferase and the oncogene *PDGF-B.* RCAS was also used to deliver the site-specific DNA recombinase Cre to delete both alleles of *Pten* that were flanked by loxP sites (termed “floxed” or FL) [19]. In addition to initiating tumorigenesis by mediating the deletion of *Pten*, Cre recombinase acted in tumor-initiating cells to delete one or both floxed alleles of *ATM*. We infected Nestin^TVA^, Pten^FL/FL^, and Atm^FL/FL^ (nPtenA ^FL/FL^) mice with the RCAS viruses to initiate brainstem tumors in which both alleles of ATM are deleted. As controls, we infected littermate Nestin^TVA^, Pten^FL/FL^, and Atm^FL/+^ (nPtenA ^FL/+^) mice with the same RCAS viruses to initiate brainstem tumors that retained expression of *ATM*.

We first examined the morphology of tumors and confirmed deletion of *Pten* in these models. Mice were sacrificed when they developed neurologic symptoms 4–12 weeks post viral injection. HE staining demonstrated expansile, hypercellular, infiltrating tumors centered in the brainstem but often extending into the adjacent thalamus and/or cerebellum (Figure 1B). Hypercellularity was observed on HE staining (Figure 1C). PDGFB was detected on the basis of an HA epitope tag, demonstrating that the hypercellular area was composed entirely of HA+ tumor cells (Figure 1D). Immunohistochemistry for PTEN demonstrated complete loss of PTEN staining within the tumor but not in surrounding normal tissues (Figure 1E). However, PTEN was present in vascular structures within the tumor (Figure 1F). These results demonstrate that in a primary mouse model, *Pten* loss combined with PDGFB can drive brainstem gliomas.

### 3.2. ATM Loss Does Not Affect Tumor Latency of Brainstem Gliomas Driven by Pten Loss

Next, we confirmed functional loss of ATM in nPtenA^FL/FL^ tumors. To control for environment and genetic background variables, these mice were compared to littermate Nestin^TVA^; Pten^FL/FL^; Atm^FL/+^ (nPtenA^FL/+^) mice, which retain a functional *Atm* allele. No differences were detected in morphology, HA staining, or PTEN staining between tumors from nPtenA^FL/+^ and nPtenA^FL/FL^ mice (Appendix A). To functionally assess deletion of *Atm*, we performed immunohistochemistry for the activated and phosphorylated form of ATM 1 h following 10 Gy of whole-brain irradiation, which elicits ATM phosphorylation [13]. A significant decrease in ATM-positive cells was observed in tumors from nPtenA^FL/FL^ mice compared to nPtenA^FL/+^ mice (Figure 2A). To confirm functional loss of ATM activity, we examined phosphorylation of the downstream ATM target KAP1 following brain irradiation [13]. A significant decrease in phospo-KAP1 staining was observed (Figure 2B). These results demonstrate functional ATM loss in tumors from nPtenA^FL/FL^ mice.

We next examined latency of nPtenA^FL/FL^ tumors compared to nPtenA^FL/+^ controls. We performed weekly in vivo luciferase imaging to detect tumors in a cohort of nPtenA^FL/FL^ and nPtenA^FL/+^ littermate-controlled mice. We examined the occurrence of a luciferase-positive signal, indicating the presence of a tumor, and sacrificed mice at humane neurologic endpoints. Tumors from both groups exhibited similar proliferation as assessed by Ki67 immunohistochemistry (Figure 2C). No significant differences were observed in the timing or penetrance of tumor detection (Figure 2D). We were unable to detect any differences in tumor size, cellular density, or other pathologic features. These results indicate that loss of ATM does not affect the latency or aggressiveness of mouse brainstem gliomas that already contain *Pten* deletion and PDGFB expression.

### 3.3. Focal Radiation Therapy Extends Survival of Mice Bearing Pten-Null Brainstem Gliomas

We next characterized a radiation therapy treatment plan that can be used to treat brainstem gliomas driven by *Pten* loss. Mice were anesthetized and the bony structures of the cranium localized by kilovoltage image guidance (Figure 3A,B). The brains were then irradiated with parallel-opposed lateral fields using a rectangular X-ray radiation field (Figure 3C). Three-dimensional volumetric planning techniques were used to achieve > 95% brainstem target coverage (Figure 3D). Since salivary gland radiation toxicity can be dose-limiting [20], we specifically optimized the plan to balance brainstem target coverage and minimization of radiation dose to the salivary glands.

We next validated an RT fractionation scheme to meet the following criteria: (i) has similar biologically effective dose [21] to clinical RT regimens for brainstem glioma; (ii) is tolerated well when delivered to the mouse brain with a focal image-guided RT platform; and (iii) balances fractionation with experimental practicality. In the clinic, brainstem glioma patients are treated with 54–60 Gy in daily fractions of 1.8–2.0 Gy. We used the biologically effective dose (BED) calculations based on the linear quadratic model to estimate tumor control and normal tissue complication rates for different experimental and clinical fractionation schemes [21,22,23,24,25,26]. We estimated a high α/β ratio of 10 for aggressive gliomas (BED10). The fractionation scheme of 10 Gy × 3 that our group used for previous brainstem glioma treatments [13] achieves BED10 = 60, which is similar to the clinical regimen of 1.8 Gy × 30 which gives a BED10 = 64 (Figure 3E).

We delivered 10 Gy × 3 to *Pten*-null brainstem glioma models and confirmed that this led to a significant increase in median overall survival, but not long-term cure (Figure 3F). The median survival of >15 weeks after radiation therapy indicates that this *Pten*-null model is relatively radiosensitive compared to other brainstem glioma genotypes [13]. These results confirm that a regimen of 10 Gy × 3 delivered with careful radiation therapy treatment planning to the brainstem can provide considerable tumor growth delay and extend mouse overall survival, without resulting in life-limiting toxicity.

### 3.4. ATM Loss Does Not Improve Survival of Pten-Null Brainstem Gliomas following Radiation Therapy

We next sought to examine whether loss of ATM can radiosensitize brainstem gliomas driven by *Pten* loss. We monitored nPtenA^FL/FL^ and nPtenA^FL/+^ until tumor formation was confirmed by luciferase imaging. Mice were then subjected to 10 Gy × 3 as detailed above. Overall survival was not significantly different for tumor-bearing nPtenA^FL/FL^ mice compared to nPtenA^FL/+^ mice (Figure 4). These results indicate that ATM loss does not radiosensitize brainstem gliomas driven by *Pten* loss.

## 4. Discussion

Here we show that loss of the tumor suppressor *Pten* can drive gliomagenesis in the mouse brainstem. Previous mouse-modeling work has used the RCAS/TVA retroviral gene delivery system to generate gliomas driven by *Pten* loss in the subventricular zone, cerebral hemisphere, and cerebellum [15]. Also, we previously incorporated *Pten* loss in brainstem tumors in addition to other drivers such as p53 deletion [16]. However, to our knowledge, primary *Pten*-null gliomas generated in the brainstem have not been previously reported. We further show that the presence or absence of a functional *Atm* allele does not affect tumor latency or aggressiveness in this *Pten*-null brainstem glioma model. Using a focal mouse brainstem irradiation approach that we characterize using a dose-volume histogram, we show that *Pten*-null brainstem gliomas are relatively sensitive to RT. However, in contrast to results obtained for primary brainstem gliomas driven by p53 loss [13], deletion of *Atm* in the tumors had no effect on mouse survival after RT. These results further support the notion that tumor genotype can impact the ability of *ATM* deletion to radiosensitize tumors and suggests that ATM inactivation does not radiosensitize brainstem gliomas driven by *Pten* loss.

ATM is a serine/threonine kinase with a conserved PI3K-like kinase. Although it is conceivable that activation of PI3K as a consequence of *Pten* deletion could compensate for loss of ATM kinase activity, our analysis of irradiated tumors with co-deletion of *Pten* and *Atm* revealed diminished phosphorylation of the ATM kinase target, KAP1 (Figure 2B). Instead, these results are more consistent with a model where the presence of functional p53 is a key determinant of radiosensitivity for brainstem gliomas in which *Atm* has been deleted. Our results for *Pten*-null mice along with our previous data for p53-null, Ink4A/ARF-null, and p53-null;Ink4A/ARF-null tumors [13] are summarized in Table 2. Similar to Ink4A/Arf-null tumors [13], *Pten*-null tumors are relatively radiosensitive at baseline, and cannot be radiosensitized further by ATM loss. Taken together, these results suggest that the presence of wild-type p53 is a key determinant as to whether primary brainstem gliomas can be radiosensitized by *Atm* deletion. Importantly, the absence of statistically significant differences does not necessarily prove non-inferiority. We therefore cannot rule out that ATM loss could have provided radiosensitization effects that our experiments were not able to detect. Ongoing work is dedicated to dissecting mechanisms of this radioresistance in p53-wild-type tumors.

An important limitation of this study was the lack of H3K27M in our brainstem glioma models. Almost all diffuse midline gliomas in the brainstem, including those with *PTEN* pathway alterations, harbor H3K27M mutations [1]. We previously showed that H3K27M represses p16 signaling to abrogate the G1/S cell-cycle checkpoint [27], which may affect radiosensitivity. H3K27M also impacts the tumor epigenetic landscape [28,29,30]. Thus, an important open question is whether ATM loss radiosensitizes brainstem gliomas of different genotypes that also harbor H3K27M. We previously introduced H3K27M to brainstem glioma mouse models via RCAS-H3K27M viral transduction [16,27,31]. For the present investigation, we similarly attempted to introduce H3K27M via RCAS-H3K27M virus transduction along with the RCAS-luciferase, RCAS-Cre, and RCAS-PDGF-B vectors (four total RCAS viruses). However, we observed variable and low levels of tumoral H3K27M expression in these attempts (data not shown). Our previous studies achieved high H3K27M expression penetrance when H3K27M was introduced alongside only RCAS-PDGF-B with or without RCAS-Cre (two to three RCAS viruses) [16,27,31]; critically, these prior studies did not rely on RCAS-luciferase. We speculate that the lower H3K27M penetrance in the present attempts may reflect diminished transduction efficiency when four or more RCAS viruses are introduced to the mouse brainstem. The critical difference that necessitated additional RCAS viruses in the present study was the need for RCAS-luciferase to facilitate tumor detection before mice became symptomatic from brainstem tumor development, to enable effective radiation therapy studies. Future work with improved complex mouse strains and/or improved viral constructs will be needed to address the effects of ATM inactivation and radiation therapy on different genotypes of brainstem gliomas containing H3K27M.

We characterize here an effective focal mouse brainstem irradiation setup and fractionation scheme that balances clinical relevance and experimental practicality. In previous work, we used a 10 Gy × 3 fractionation scheme using a similar beam arrangement to extend the survival of mice bearing brainstem gliomas [13,32]. To our knowledge this regimen had not previously been volumetrically characterized using 3D radiation treatment planning techniques, or tested in a primary *Pten*-null brainstem glioma model. In the current work, we use volumetric planning techniques to define dose-volume relationships for this radiation plan. A key feature of the current radiation plan is the ability to spare the parotid and submandibular glands while delivering a full radiation dose to most of the brainstem target. BED10 estimates show that this 10 Gy × 3 regimen provides a comparable BED10 to the 6-week 54 Gy/30 regimens frequently used in the clinic. In contrast, other commonly used fractionation schemes for experimental mouse brain treatment such as 2 Gy × 5, 5 Gy × 5, and 15 Gy × 1 exhibit BED10 considerably lower than 50 (Figure 3E). Importantly, the BED10 calculations do not account for total treatment time, which has been demonstrated to be a key factor in predicting tissue response to injury [33]. Thus, the time-dependent biologically effective dose may be substantially higher in the 10 Gy × 3 fractionation scheme than that seen in patients, which would explain the dramatic treatment effects seen in this study compared to the outcomes in humans with brainstem gliomas. We observed that 10 Gy × 3 cannot be delivered to the mouse brain using plans that do not spare the salivary glands without life-limiting toxicity. This is likely because doses above ~10–15 Gy in a single fraction to the entire salivary apparatus are associated with dose-limiting lethal toxicity [20]. We propose that a critical factor to achieve clinically relevant effective doses in experimental mouse brainstem irradiation is the ability to spare the salivary glands from dose-limiting toxicity using image guidance and a lateral beam configuration.

## 5. Conclusions

Our results highlight the importance of defining tumor genotypes in pediatric brainstem gliomas when investigational therapeutics are evaluated. For example, clinical trials such as PNOC008 (NCT03739372) seek to match pediatric brain tumor patients with investigational therapies based on tumor genotype. As ATM inhibitors have now entered clinical trials with radiation therapy for adults with brain tumors or brain metastases (NCT03423628), our results suggest correlation of *TP53* status and *PTEN* pathway mutational status will be important to include in any future clinical trials of ATM inhibitors with radiation therapy in brainstem gliomas. If ATM inhibitors do not radiosensitize *TP53*-wild-type tumors as predicted by our studies, then the primary brainstem gliomas driven by *PTEN* pathway alterations with wild-type *TP53* can be used to search for alternative therapeutic approaches to radiosensitize tumors with this genotype.

## Figures and Tables

**Table 2 cancers-14-04506-t002:** Summary of the effects of ATM loss on radiosensitivity in primary mouse brainstem glioma models of different genotypes.

Mouse Strain	p53	Ink4A/Arf	Pten	Baseline Radiosensitivity (Atm^FL/+^)	Effect of ATM Loss on Radiosensitivity (Atm^FL/FL^)	Ref.
nPA	-	+	+	Resistant	Radiosensitization	[13]
nIA	+	-	+	Sensitive	None	[13]
nPIA	-	-	+	Very resistant	Radiosensitization	[13]
nPtenA	+	+	-	Sensitive	None	This work

Abbreviations: nPA = Nestin^TVA^; p53^FL/FL^; Atm^FL/+^ or Atm^FL/FL^; nIA = Nestin^TVA^; Ink4A/Arf^FL/FL^; Atm^FL/+^ or Atm^FL/FL^; nPIA = Nestin^TVA^; p53^FL/FL^; Ink4A/Arf^FL/FL^; Atm^FL/+^ or Atm^FL/FL^; nPtenA = Nestin^TVA^; Pten^FL/FL^; Atm^FL/+^ or Atm^FL/FL^.

## Data Availability

Not applicable.

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
