# Peer review of "The Effect of Atm Loss on Radiosensitivity of a Primary Mouse Model of Pten-Deleted Brainstem Glioma"

_cancers, 2022, doi:10.3390/cancers14184506_

Round 1

Reviewer 1 Report

Stewart et al. present a study using the RCAS-TVA system of gliomagenesis, which utilizes injection of chicken fibroblasts producing viruses containing various inducible expression constructs into Cre-Lox’d mice to achieve site-selective activation of select molecular pathways. The authors have extensively characterized this model in previous publications. Several authors on this manuscript recently published their findings using this model that ATM loss slows tumor latency and improves overall survival in TP53-KO but not TP53-wt mice (Deland et al 2021). The study presented in the current manuscript seeks to build on these findings by asking whether loss of PTEN can alter the radiosensitizing effect of ATM loss in TP53-wt tumors in this model? 

In short, the authors find ATM status does not alter RT sensitivity in the background of PTEN mutation, and that PTEN-mutant tumors remain radiosensitive in their model. Overall, the results of this study may be clinically meaningful in that they further demonstrate TP53 mutation status is a critical determinant of radiation response and ATM inhibitor sensitivity. The manuscript is well-written. I have no concerns regarding conflicts of interest. 

I have a few comments on the study:

1. The authors preface the manuscript by describing PTEN (and associated pathway) mutations in pediatric brainstem gliomas, of which the majority contain H3K27M mutations. However, the transformation system described does not recapitulate this key event in brainstem gliomagenesis. A previous study performed by some authors on this manuscript showed RCAS-TVA-mediated expression H3.3K27M accelerated gliomagenesis in a PDGF-B-over-expressing, TP53-wt background (Cordero et al 2017). The data presented in the paper therefore model a very rare type of brain tumor (H3-wildtype, PTEN-mutant DMG; N=0-1 in Mackay et al 2017). The authors briefly comment on this in the Discussion (line 361), but I think this is a critical molecular feature to include in this study. Can the authors comment on the rationale for not including this genetic alteration in their study design? The statements regarding the clinical application of the results should be moderated given the limited translational scope in this regard. 

2. The authors take some liberty in claiming their RT model (3 x 10 Gy) recapitulates the same biologically effective dose seen by patients (1.8-2Gy x 54). While the authors use the biologically effective dose (BED) calculation proposed by Fowler et al, this calculation does not take into account treatment time, which other definitions of the biological effective dose have demonstrated to be a key factor in predicting tissue response to injury (as calculated in PMID 29745754). My suspicion is the time-dependent BED is substantially higher in this model than that seen in patients, which would explain the dramatic treatment effects seen in this study which are not observed in humans with DMG.  

3. Lines 297-302: Please moderate the statements that the radiotherapy results accurately recapitulate clinical observations following RT for DMG, which is effectively palliative as currently administered. 

Author Response

Stewart et al. present a study using the RCAS-TVA system of gliomagenesis, which utilizes injection of chicken fibroblasts producing viruses containing various inducible expression constructs into Cre-Lox’d mice to achieve site-selective activation of select molecular pathways. The authors have extensively characterized this model in previous publications. Several authors on this manuscript recently published their findings using this model that ATM loss slows tumor latency and improves overall survival in TP53-KO but not TP53-wt mice (Deland et al 2021). The study presented in the current manuscript seeks to build on these findings by asking whether loss of PTEN can alter the radiosensitizing effect of ATM loss in TP53-wt tumors in this model? 

In short, the authors find ATM status does not alter RT sensitivity in the background of PTEN mutation, and that PTEN-mutant tumors remain radiosensitive in their model. Overall, the results of this study may be clinically meaningful in that they further demonstrate TP53 mutation status is a critical determinant of radiation response and ATM inhibitor sensitivity. The manuscript is well-written. I have no concerns regarding conflicts of interest. 

Thank you for taking the time to review our manuscript and for the helpful feedback.

I have a few comments on the study:

  1. The authors preface the manuscript by describing PTEN (and associated pathway) mutations in pediatric brainstem gliomas, of which the majority contain H3K27M mutations. However, the transformation system described does not recapitulate this key event in brainstem gliomagenesis. A previous study performed by some authors on this manuscript showed RCAS-TVA-mediated expression H3.3K27M accelerated gliomagenesis in a PDGF-B-over-expressing, TP53-wt background (Cordero et al 2017). The data presented in the paper therefore model a very rare type of brain tumor (H3-wildtype, PTEN-mutant DMG; N=0-1 in Mackay et al 2017). The authors briefly comment on this in the Discussion (line 361), but I think this is a critical molecular feature to include in this study. Can the authors comment on the rationale for not including this genetic alteration in their study design? The statements regarding the clinical application of the results should be moderated given the limited translational scope in this regard. 

We agree with this comment and have added this paragraph to the discussion (please see lines 365-387 of the revised manuscript, which are also copied below):

An important limitation of this study was the lack of H3K27M in our brainstem glioma models.  Almost all diffuse midline gliomas in the brainstem, including those with PTEN pathway alterations, harbor H3K27M mutations [1].  We previously showed that H3K27M represses p16 signaling to abrogate the G1/S cell cycle checkpoint [27], which may affect radiosensitivity.  H3K27M also impacts the tumor epigenetic landscape [28-30].  Thus, an important open question is whether Atm loss radiosensitizes brainstem gliomas of different genotypes that also harbor H3K27M.  We previously introduced H3K27M to brainstem glioma mouse models via RCAS-H3K27M viral transduction [16, 27, 31].  For the present investigation, we similarly attempted to introduce H3K27M via RCAS-H3K27M virus transduction along with the RCAS-luciferase, RCAS-Cre, and RCAS-PDGF-B vectors (four total RCAS viruses).  However, we observed variable and low levels of tumoral H3K27M expression in these attempts.  Our previous studies achieved high H3K27M expression penetrance when H3K27M was introduced alongside only RCAS-PDGF-B with or without RCAS-Cre (two to three RCAS viruses) [16, 27, 31]; critically these prior studies did not rely on RCAS-luciferase.  We speculate that the lower H3K27M penetrance in the present study may reflect diminished transduction efficiency when four or more RCAS viruses are introduced to the mouse brainstem.  The critical difference that necessitated additional RCAS viruses in the present study was the need for RCAS-luciferase to facilitate tumor detection before mice became symptomatic from brainstem tumor development, to enable effective radiation therapy studies.  Future work with improved complex mouse strains and/or improved viral constructs will be needed to address the effects of ATM inactivation and radiation therapy on different genotypes of brainstem gliomas containing H3K27M.

  1. The authors take some liberty in claiming their RT model (3 x 10 Gy) recapitulates the same biologically effective dose seen by patients (1.8-2Gy x 54). While the authors use the biologically effective dose (BED) calculation proposed by Fowler et al, this calculation does not take into account treatment time, which other definitions of the biological effective dose have demonstrated to be a key factor in predicting tissue response to injury (as calculated in PMID 29745754). My suspicion is the time-dependent BED is substantially higher in this model than that seen in patients, which would explain the dramatic treatment effects seen in this study which are not observed in humans with DMG.  

Thank you for this important insight.  We have added this to the discussion as follows:

“Importantly, the BED10 calculations do not account for total treatment time, which has been demonstrated to be a key factor in predicting tissue response to injury [30].  Thus, the time-dependent biologically effective dose may be substantially higher in the 10 Gy x 3 fractionation scheme than that seen in patients, which would explain the dramatic treatments effects seen in this study compared to the outcomes in humans with brainstem gliomas.”

  1. Lines 297-302: Please moderate the statements that the radiotherapy results accurately recapitulate clinical observations following RT for DMG, which is effectively palliative as currently administered. 

We have now removed the statements about the mouse radiotherapy recapitulating clinical observations from this paragraph.

Reviewer 2 Report

Thank a lot for gave me the possibility to revise manuscript.

The idea is attractive and the potential impact on clinical trial is significant.

Further studies that will be focus on the correlation with H3K27 status as well as with other genetic and epigenetic hallmarks are needed to best define the radiosensitivity of brainstem gliomas.

The way taken by the authors is complex but promising.

The paper is suitable for pubblication.

Author Response

Thank you for taking the time to consider our manuscript and for the positive feedback.

Further studies that will be focus on the correlation with H3K27 status as well as with other genetic and epigenetic hallmarks are needed to best define the radiosensitivity of brainstem gliomas.

We agree with this important point.  As detailed in the response to Reviewer 1, we have added a paragraph considering the important correlation with H3K27M and other genetic and epigenetic hallmarks to best define radiosensitivity of brainstem gliomas (please see lines 365-387 of the revised manuscript, which are also copied below).

An important limitation of this study was the lack of H3K27M in our brainstem glioma models.  Almost all diffuse midline gliomas in the brainstem, including those with PTEN pathway alterations, harbor H3K27M mutations [1].  We previously showed that H3K27M represses p16 signaling to abrogate the G1/S cell cycle checkpoint [27], which may affect radiosensitivity.  H3K27M also impacts the tumor epigenetic landscape [28-30].  Thus, an important open question is whether Atm loss radiosensitizes brainstem gliomas of different genotypes that also harbor H3K27M.  We previously introduced H3K27M to brainstem glioma mouse models via RCAS-H3K27M viral transduction [16, 27, 31].  For the present investigation, we similarly attempted to introduce H3K27M via RCAS-H3K27M virus transduction along with the RCAS-luciferase, RCAS-Cre, and RCAS-PDGF-B vectors (four total RCAS viruses).  However, we observed variable and low levels of tumoral H3K27M expression in these attempts.  Our previous studies achieved high H3K27M expression penetrance when H3K27M was introduced alongside only RCAS-PDGF-B with or without RCAS-Cre (two to three RCAS viruses) [16, 27, 31]; critically these prior studies did not rely on RCAS-luciferase.  We speculate that the lower H3K27M penetrance in the present study may reflect diminished transduction efficiency when four or more RCAS viruses are introduced to the mouse brainstem.  The critical difference that necessitated additional RCAS viruses in the present study was the need for RCAS-luciferase to facilitate tumor detection before mice became symptomatic from brainstem tumor development, to enable effective radiation therapy studies.  Future work with improved complex mouse strains and/or improved viral constructs will be needed to address the effects of ATM inactivation and radiation therapy on different genotypes of brainstem gliomas containing H3K27M.

Reviewer 3 Report

In this work, authors complement the findings of a previous report (ref 13) in which it was shown that knocking out Atm on p53 deficient mice enhanced tumour radiosensitivity. Here, they demonstrate that ATM and PTEN codeletion does not affect survival after radiotherapy, using basically the same approach of ref 13. In this sense, table 1 at the end clarifies a lot of the results, as much of this work must be understood with regards to reference 13. 

The importance of this work relies in the fact that ATM inhibitors are entering in clinical trials.

This reviewer has a small request: A table with the number of mice undergoing the different tests would be helpful for readers. For example this reviewer was curious about the immunochemistry for the phosporilation of ATM one hour after therapy. The mice were sacrified I guess, and then other unmatched animals had to be used for the tumor-free survival calculation, right? This might seem obvious for the authors, but not for a reader, after reading. Survival is calculated for about 50 animals each group, but then survival curves for radiation are from less than 20 animals at each group. A little table would help readers to see how experiments were done and to appreciate the total number of mice employed. 

Author Response

In this work, authors complement the findings of a previous report (ref 13) in which it was shown that knocking out Atm on p53 deficient mice enhanced tumour radiosensitivity. Here, they demonstrate that ATM and PTEN codeletion does not affect survival after radiotherapy, using basically the same approach of ref 13. In this sense, table 1 at the end clarifies a lot of the results, as much of this work must be understood with regards to reference 13. 

The importance of this work relies in the fact that ATM inhibitors are entering in clinical trials.

Thank you for taking the time to consider our manuscript and for the positive feedback.

This reviewer has a small request: A table with the number of mice undergoing the different tests would be helpful for readers. For example this reviewer was curious about the immunochemistry for the phosporilation of ATM one hour after therapy. The mice were sacrified I guess, and then other unmatched animals had to be used for the tumor-free survival calculation, right? This might seem obvious for the authors, but not for a reader, after reading. Survival is calculated for about 50 animals each group, but then survival curves for radiation are from less than 20 animals at each group. A little table would help readers to see how experiments were done and to appreciate the total number of mice employed. 

We thank the reviewer for this helpful suggestion to clarify the work.  We now provide this as Table 1 in the Methods.

Round 2

Reviewer 1 Report

Thank you for incorporating a discussion of the limitations of the study in regard to inclusion of an H3K27M-mutant strain and the translational validity of the radiotherapy model described in the study. 

As the addition of a fourth RCAS construct is not technically feasible in the model described, additional edits are necessary to inform readers that H3-mutant tumors are not within the translational scope of the manuscript, as this is currently not readily apparent and will not be assumed by most readers at first glance. To prevent this misunderstanding, please add "H3-wildtype" as a modifier to all instances of "brainstem glioma" in the title, summary and abstract to prevent readers from misinterpreting the scope of the study.